# Knowledge, Attitudes, and Common Practices of Livestock and Poultry Veterinary Practitioners Regarding the AMU and AMR in Bangladesh

**DOI:** 10.3390/antibiotics11010080

**Published:** 2022-01-10

**Authors:** Md. Abul Kalam, Md. Sahidur Rahman, Md. Abdul Alim, Shahanaj Shano, Sharmin Afrose, Faruk Ahmed Jalal, Samira Akter, Shahneaz Ali Khan, Md. Mazharul Islam, Md Bashir Uddin, Ariful Islam, Ricardo J. Soares Magalhães, Mohammad Mahmudul Hassan

**Affiliations:** 1Bangladesh Country Office, Helen Keller International, Dhaka 1212, Bangladesh; a.kalam724@gmail.com; 2One Health Center for Research and Action, Akbarshah, Chattogram 4207, Bangladesh; sahid.dvm@gmail.com; 3Faculty of Veterinary Medicine, Chattogram Veterinary and Animal Sciences University, Khulshi, Chattogram 4225, Bangladesh; maalim85@gmail.com (M.A.A.); shahneazbat@gmail.com (S.A.K.); 4Institute of Epidemiology, Disease Control and Research, Dhaka 1212, Bangladesh; shahanajshano@gmail.com (S.S.); arif@ecohealthalliance.org (A.I.); 5EcoHealth Alliance, New York, NY 10018, USA; 6Bangladesh Country Office, World Food Programme, Dhaka 1212, Bangladesh; sharminafrosenupur@gmail.com; 7Handicap International-Humanity & Inclusion, Dhaka 1212, Bangladesh; farukahmedjalal@gmail.com; 8Department of Anthropology, Jahangirnagar University, Dhaka District, Savar 1212, Bangladesh; samira68anth@gmail.com; 9Department of Animal Resources, Ministry of Municipality and Environment, Doha P.O. Box 35081, Qatar; mmmohammed@mme.gov.qa; 10Department of Medicine, Sylhet Agricultural University, Sylhet 3100, Bangladesh; bashir.vetmed@sau.ac.bd; 11Centre for Integrative Ecology, Geelong Campus, School of Life and Environmental Science, Deakin University, Warrnambool, VIC 3216, Australia; 12UQ Spatial Epidemiology Laboratory, School of Veterinary Science, The University of Queensland, Gatton, QLD 4343, Australia; 13UQ Child Health Research Centre, Children’s Health and Environment Program, The University of Queensland, South Brisbane, QLD 4101, Australia

**Keywords:** antimicrobials, resistance, prescription behavior, factors, veterinarians, Bangladesh

## Abstract

Current evidence indicates that more than half of all antimicrobials are used in the animal food-producing sector, which is considered a significant risk factor for the development, spread, and existence of antimicrobial resistance (AMR) pathogens in animals, humans, and the environment. Among other factors, clinical etiology and the level of knowledge, attitudes, and practices (KAP) of veterinarians are thought to be responsible for inappropriate prescriptions in the animal-source protein production sector in lower-resource settings like Bangladesh. We performed this cross-sectional study to assess factors associated with veterinarians’ antimicrobial prescription behavior and their KAP on antimicrobial use (AMU) and AMR in Bangladesh. Exploratory and multivariate logistic models were used to describe an association between knowledge, attitudes, and practices of AMU and AMR and demographic characteristics of veterinarians. The results demonstrated that when selecting an antimicrobial, there was no to minimal influence of culture and susceptibility tests and patients’ AMU history but moderate to high influence of the farmer’s economic condition and drug instructions among the veterinarians. The results also demonstrated that more than half of the veterinarians had correct KAP regarding AMU and AMR, while the rest had moderate or lower levels of KAP. The factor score analysis revealed that age, level of education, years of experience, gender, and previous training on AMU and AMR were the key influencing factors in their level of KAP. Adjusted logistic regression analysis showed that respondents’ age, current workplace, and previous training on AMU and AMR had a positive association with increased KAP. Considering the results, it is imperative to include AMR issues on vet curricula, and to provide post-education training, awareness campaigns, easy access to, and dissemination of AMR resources. Increasing the veterinary services to the outreach areas of the country and motivating veterinarians to follow the national AMR guidelines could be some other potential solutions to tackle the over-prescriptions of antimicrobials.

## 1. Introduction

Antimicrobial resistance (AMR) is a cross-disciplinary global public health threat for human, animal, and environmental health [1,2,3,4,5]. Globally, more than half of antimicrobials are used in food-producing animals [6,7], which is almost four times more than in the human sector. The inappropriate use of antimicrobials in industrial animal farming can constitute a risk factor for the development and spread of AMR in pathogens [8,9,10]. The international literature has shown a significant positive correlation between inappropriate antimicrobial use and the development of resistance [11,12,13]. Resistant pathogens can be transmitted from animals to humans and the environment by direct exposure, residue in foods [14,15,16], and contaminated farm wastes [17,18,19]. The direct effects of AMR on the animal sector are production losses, increased treatment costs, and hampered food safety [20,21]. It is estimated that around two million AMR infections occur, with 23,000 deaths in humans each year in the USA alone [22,23], and the number is 700,000 worldwide [24]. The use of antimicrobials in food animals has increased suddenly and is predicted to reach 82% by 2030 in Asian countries [25], and Bangladesh is no exception. Multi-drug resistant (MDR) *E. coli*, *Salmonella* spp., *Enterobacter* spp., *Staphylococcus aureus*, *Pasteurella* spp., *Bacillus* spp., and *Campylobacter* spp. have been isolated from Bangladeshi broilers, layers, day-old chicks, pigeons, frozen chickens, and the milk of mastitis-affected cows [26,27,28,29,30,31]. This evidence suggests that Bangladesh is currently living through an emerging AMR problem in animal-source protein production sectors such as poultry, eggs, milk, and meat. Part of the problem lies in the largely unregulated access to, and delivery of, antimicrobial products. For example, poultry farmers in Bangladesh are bound to local feed and drug sellers who enable indiscriminate access to antibiotics to commercial poultry farmers, who then use these products on their own schedules largely without veterinary supervision [32]. A study stated that one third of commercial poultry farmers use antimicrobials given to them by different non-vet prescriber groups such as dealers, local expert farmers or themselves [33].

In Bangladesh, inappropriate use of antimicrobials has led to the development of different pathogenic and zoonotic AMR microbes in the livestock sector [20]. This creates an increasing political pressure to implement the proper strategies, and consequently, Bangladesh has adopted the National Action Plan (2017–2022) for the containment of AMR [34]. However, there are distinct policy and implementation gaps in the veterinary sector, such as the inclusion of AMR in the veterinary curriculum and proper prescribing guidelines for appropriate use of antimicrobials in the livestock sector [35]. The prudent use of antimicrobials is crucial to prevent resistance development, and this can be achieved by introducing changes in prescribing behavior and the AMR KAP of prescribers [36]. A study in the Netherlands found that veterinarians with favorable attitudes towards the prudent use of antimicrobials were positively affecting their farmer clients and, as a result, reducing AMR [37]. In on-field practices, veterinarians are thought to be countered by several clinical and non-clinical factors such as farmers’ or owners’ demand for antimicrobials and the business policies of companies [38,39]. Veterinarians’ prescribing varied depending on the farmers’ perception of antimicrobials, farm biosecurity practice, and socio-economic conditions [40,41,42]. Considering this context, the prescribing decisions of veterinarians mostly depend on their knowledge of antimicrobial resistance and attitudes on adherence to guidelines [20]. Moreover, the lack of proper monitoring, poor biosecurity in the animal healthcare system, unqualified informal prescriber groups, and laboratory diagnosis facilities stand as barriers to antimicrobial stewardship in the veterinary sector in Bangladesh. Active involvement of veterinarians in the AMU is the key step for reducing the prevalence of AMR. Moreover, studies on the knowledge gaps, negative attitudes, and practices with respect to antimicrobial prescribing among practicing veterinarians in Bangladesh are limited. To address this gap in knowledge directly, we conducted this cross-sectional survey to assess the level of knowledge, attitudes, and common practices of livestock and poultry veterinary practitioners regarding the AMU and AMR.

## 2. Materials and Methods

### 2.1. Study Design, Population, and Recruitment

This cross-sectional survey was carried out from April 2021 to May 2021 to investigate prescription behaviors regarding antibiotic use among vets from different areas in Bangladesh. Our target participants were livestock and poultry veterinarians who were actively engaged in veterinary practices during the study. We excluded pet practitioners, as we focused on antimicrobial resistance in food producing animals. A Google-Form-based self-administered questionnaire was designed, and a web link was distributed to veterinarians via different professional associations, personal email, university groups, social media sites, and websites.

### 2.2. Questionnaire Development

A questionnaire was developed based on a comprehensive review of the published literature in order to identify factors influencing antimicrobial prescribing behaviors and knowledge, attitudes, and practices (KAP) regarding AMU and AMR among veterinarians. The questionnaire was divided into six sections: (i) personal information; (ii) the sources of information on AMR and AMU; (iii) clinical etiological factors influencing antimicrobial selection; (iv) knowledge factors; (v) attitude factors; and (vi) practice factors regarding antimicrobial prescriptions. In the personal information section, we asked about the respondents’ age, gender, level of education, years of experience in practice, area of practice, nature of the workplace, and any training on AMR and AMU in the last year. The second section consisted of one multi-response question to capture the sources they used to obtain information on AMU and AMR. We asked ten questions (three negative and seven positive) to understand the factors that influence antimicrobial prescription decisions in the third section. Twelve questions were asked to assess the knowledge factors on AMR and AMU in the fourth section, and the measurement of each question was determined based on the self-reported responses from “good” to “no knowledge at all”. To assess the attitude factors, in the fifth section, we used a four-point Likert scale (from “strongly agree” to “strongly disagree”) and asked ten questions (three negative and seven positive). The final section consisted of twelve questions (six negative and six positive) to assess the practice factors on AMU and AMR.

The preliminary draft of the questionnaire was reviewed by three expert researchers to identify ambiguity and assess content validity. After obtaining and incorporating expert feedback, the questionnaire was tested among ten respondents to check the language suitability and the appropriateness of the questions. Slight modifications of language were recorded during the pilot phase, and these modifications were addressed before data collection. The pilot responses were excluded from the current analysis.

### 2.3. Sampling Procedure

Before inviting the veterinarians to participate in the study, we collected a list of veterinarians (*n* = 5800) from the Bangladesh Veterinary Council. Similarly, to include the intern veterinarians, we collected the list of interns working in different veterinary hospitals in different Upazilas in Bangladesh. The study population included veterinarians who fulfilled the inclusion criteria of being a practitioner in either commercial poultry, livestock, or both sectors. The sample size was calculated using the Raosoft calculator (Raosoft: http://www.raosoft.com/samplesize.html?nosurvey accessed on 12 February 2021). A sample size of 377 was estimated based on a 50% response distribution, a 5% margin of error, and a 95% confidence interval. The expected response proportion of 50% was assumed because both responses and response rates were completely unknown since there are no similar previously published studies from Bangladesh. We invited 500 vets to participate in the current study and obtained a response from 436.

### 2.4. Ethical Statement

The study was conducted by following the Declaration of Helsinki, and the protocol was approved by the Ethics Committee of the Chattogram Veterinary and Animal Sciences University, Bangladesh (permit reference number: CVASU/Dir (R and E) EC/2019/126 (02), Date: 29 December 2019). The details of the participants were anonymous, and data confidentiality was properly maintained (see Appendix A). The nature of the study was completely voluntary, and consent was obtained appropriately from all the subjects before their inclusion in the study.

### 2.5. Statistical Analysis

The completed questionnaires were manually checked for data quality before coding using Microsoft^®^ Office Excel 2010 (Appendix A). Cronbach’s alpha was used to measure the internal consistency of the questionnaire. The reproducibility was evaluated using intra-class correlations for each section of the questionnaire, with an acceptable value being ≥0.82. The calculation for Cronbach’s alpha found as 0.77 for prescription factors, 0.87 for knowledge questions, 0.68 for attitudes questions, and 0.77 for the practice theme.

A four-point index (composite score range: 0 to 3) was assigned to responses of “no influence” to “high influence” for factors influencing prescription decisions. Similar index values were used for responses “not at all” to “good” for knowledge questions, “strongly disagree” to “strongly agree” for attitudes questions, and “never” to “regularly” for practice questions. To analyze how individual participants performed overall in the knowledge, attitude, and practice categories, the sum of each participant’s answers for that particular section was calculated. Data were analyzed using the statistical software Stata/SE 16.1 (StataCorp, 4905, Lake Way Drive, College Station, TX, USA). We used descriptive statistics such as frequencies and percentages. Relationships between independent samples were explored using the chi-square test to determine if there were differences among respondents’ characteristics with respect to the themes. Using the principal factor method described, we identified significant factors in the demographic characteristics and themes.

Furthermore, this factor score analysis was used as a part of the adjusted multivariable logistic regression analysis to determine the associations between the knowledge, attitudes, and practice themes and the respondents’ demographics. The outcome variables regarding knowledge, attitudes, and practices were categorized as “incorrect”, “moderate”, and “correct”; “unfavorable”, “moderate”, and “favorable”; and “bad”, “moderate”, and “good”, respectively. In doing so, we constructed a three-point index (composite score range: 0 to 2) and assigned values to responses for knowledge, attitude, and practice items. For knowledge questions, we assigned 2 for a correct response, 1 for moderate, and 0 for an incorrect value. Similarly, for attitude questions, we assigned 2 for favorable, 1 for moderate, and 0 for unfavorable attitudes. The same strategy was applied for practice questions, where the categories were good (2), moderate (1), and bad (0). Before categorizing, the negative items were reversed and calculated accordingly. These outcome variables were then correlated with the explanatory variables. The results were expressed as odds ratios (ORs) accompanied by 95% confidence intervals (95% CIs), and a *p*-value of <0.05 was used as the threshold for statistical significance.

## 3. Results

### 3.1. Demographic and Socio-economic Characteristics of Respondents

Among the veterinary respondents, 83.3% (*n* = 363) were male and 16.7% (*n* = 73) were female (Table 1).

While close to 40% (*n* = 168) of respondents were veterinary interns, 31% (*n* = 134) and 19% (*n* = 84) of respondents were with up to 3 years of work experience and 7 years or more experience, respectively. Most of the respondents (*n* = 235) were from the government sector (Department of Livestock Services), followed by the private sector (*n* = 136) and medicine/feed companies (*n* = 65). More than half of respondents (56%, *n* = 246) had not received any training on antimicrobial use and AMR at the time of the current study.

### 3.2. Sources of Information on AMU and AMR

Figure 1 shows the sources that respondents used in order to seek information on AMU and AMR. Among the different options, the top five information sources were previous knowledge of AMU or training (*n* = 350), senior colleagues (*n* = 348), the experience of managing similar problems (307), use of the Internet (*n* = 235), and national guidelines and protocols on AMR and AMU (*n* = 167).

### 3.3. Role of Clinical Etiological and Other Factors Influencing the Selection of Appropriate Antimicrobials

Selection of appropriate antimicrobials by a veterinary respondent was influenced mostly by disease type/organism (mentioned by 71% of all respondents) followed by clinical sign (61%) and patients’ previous history of AMU and the potential side effects of antimicrobials (59%) (Figure 2). Interestingly, around 41% acknowledged that there was no to minimal influence of culture and susceptibility testing in their antimicrobial prescribing practice. On the other hand, guidelines/drug instruction, the economic status of the owner, and the route of administration were moderately influenced (42%, 44%, and 41%, respectively) on their antimicrobial prescribing decision.

### 3.4. Knowledge Factors in Prescribing Antimicrobials

Overall, our results show that respondents had relatively good knowledge of AMR and AMU (Table 2). In particular, respondents indicated good knowledge of different classes and generations of antibiotics (63%, *n* = 175), choosing the correct antimicrobials (52%, *n* = 227), correct doses (65%, *n* = 282), correct routes for administering antimicrobials (75%, *n* = 327), duration of antimicrobial treatments (51%, *n* = 223), and causes of AMR (60%, *n* = 263). Almost half of all respondents had a medium level of knowledge on interpreting laboratory results (50%, *n* = 217), more than half (56%) reported knowledge on using a combination of different antimicrobials, and nearly half of respondents (48%) had knowledge on modifying or stopping the use antimicrobials (48%, *n* = 210). The majority of respondents had a medium-to-poor levels of knowledge of reserve groups of antibiotics (around 56%), knowledge of critically important antimicrobials (around 58%), and knowledge of the national action plan for AMR (around 66%).

### 3.5. Attitude Factors

Our analysis indicates that most veterinary respondents (*n* = 384) strongly agreed that there is a threat of antimicrobial resistance for livestock and poultry production (Table 3). Most of them also agreed (*n* = 170) and strongly agreed (*n* = 164) with the statement that “a single course of antibiotics can cause antimicrobial resistance” whereas 23% (*n* = 102) disagreed with this statement. Moreover, 61% (*n* = 265) of participants strongly agreed with the statement that “irrational antibiotic uses in animals lead to antibiotic resistance in humans”. More than 70% (*n* = 322) of participants also agreed that AMR is a natural as well as an anthropogenic phenomenon, while one quarter of them (26%, *n* = 114) disagreed with this statement.

More than 80% of participants (*n* = 428) strongly agreed that antimicrobial resistance will become more problematic in the near future. In contrast, most of the respondents (77%, *n* = 336) believed (agreed and strongly agreed) that new antimicrobials will be developed to tackle the AMR issue. However, almost 80% (*n* = 349) of all respondents agreed that they faced difficulties when selecting the correct antimicrobials and the same number of participants (*n* = 349) also agreed that they had enough sources of information on antimicrobials and their usage.

Although 81% (*n* = 352) of respondents agreed and strongly agreed with restricting priority antibiotics for human use only, almost 20% (*n* = 84) of all respondents either disagreed or strongly disagreed with the idea of imposing this restriction.

### 3.6. Practice Factors on AMU and AMR

Most respondents indicated that they regularly give advice to their farmers about the withdrawal period of the antimicrobials they prescribed, keeping records of antimicrobials that are used (47%, *n* = 206), completing the full course of antimicrobials (81%, *n* = 352), and proper vaccination of animals to reduce the use of antimicrobials (76%, *n* = 330) (Table 4). However, almost half of respondents (49%) regularly or frequently prescribed antimicrobials through telephone conversations with farmers, which is not ideal. They also used antibiotics for prophylaxis regularly (14.5%), frequently (37%), or rarely (35%). Interestingly, around 40% of all respondents prescribed more than one antimicrobial regularly or frequently in a single prescription. Around 36% of respondents acknowledged that they either frequently or regularly used antimicrobials due to the demand of farmers. Respondents also used a higher dose of antimicrobials for rapid recovery of their patients regularly (11.2%, *n* = 49) or frequently (27.5%, *n* = 120).

The highest number of respondents did not follow culture and susceptibility (CS) testing for selecting the right antibiotics, and 26.4% (*n* = 115) of them never used the CS test.

### 3.7. Associations with the Level of Antimicrobial Knowledge, Attitudes, and Practices

The analysis further revealed that respondents’ age (*p* = 0.003), level of education (*p* = 0.031), years of experience in practice (*p* = 0.018), and previous training on AMU and AMR (*p* = 0.000) were significant factors affecting their knowledge (Table 5). In terms of their practices, gender (*p* = 0.016) and level of education (*p* = 0.027) were the factors affecting their practices.

Our adjusted logistic regression analysis showed that respondents’ current workplaces and previous training on AMU and AMR were positively associated with increased levels of knowledge on AMU and AMR (Table 6). Specifically, veterinarians who worked in government hospitals were 2.09 times more likely to have a higher AMU and AMR knowledge score (OR = 2.09, CI = 1.06–4.10, *p* = 0.032) than veterinarians who worked in medicine/feed companies. Moreover, the analysis also found that respondents who received training on AMU and AMR were 1.92 times more likely to have higher AMU and AMR knowledge (OR = 1.92, CI = 1.23–2.97, *p* = 0.004) compared to those who did not receive training.

The analysis further revealed that the respondents’ age and previous training were positively associated with favorable attitudes towards AMU and AMR. The respondents who were 36–40 years of age were 0.24 times more likely to have favorable attitudes towards AMU and AMR (OR = 0.24, CI = 0.06–0.97, *p* = 0.043) compared to those who were 18–25 years of age. Further, the respondents who received training on AMU and AMR were 2.09 times more likely to have favorable attitudes (OR = 2.09, CI = 1.35–3.25, *p* = 0.001) than those who had not received training.

Like the knowledge and attitude themes, training was positively associated with prescription practices. The respondents who received training on AMU and AMR were 0.76 times more likely to perform well when writing a prescription (OR = 0.76, CI = 0.50–1.16, *p* = 0.024) than those who had not received training.

## 4. Discussion

Antimicrobial resistance (AMR) has become a burgeoning public health issue globally, including in Bangladesh. AMR is aggravated by many factors, largely due to the over-use of antimicrobials (AMU) and unregulated diverse health systems [28]. The reduction of AMR in the animal industries sector requires intervention from all stakeholders (e.g., veterinary students, para-vets, drug and feed sellers, and farmers), and especially from veterinarians. They are considered the key players in changing the prescribing behavior of other stakeholders such as drug and feed sellers, para-vets, and commercial farmers [32,33,43]. To the best of our knowledge, this is the first study of its type undertaken in Bangladesh assessing the factors associated with veterinarians’ prescription behaviors and their knowledge, attitudes, and practices.

Our study demonstrated that prescription of antimicrobials was influenced by several factors including diseases or organisms, history of clinical signs, and potential side effects of the antimicrobials, which were good practices. Overall, the factors associated with the selection of antimicrobials by the veterinarians were good. Previous reports also observed that experience through appropriate training, access to published literature, and availability of the treatment guidelines has a significant role in changing the prescription behavior of veterinarians [43,44]. Giving priority to the types of diseases or organisms in selecting or prescribing the antimicrobials is an excellent practice observed in this study, which prevents the misuse of antimicrobials. Unfortunately, it is a matter of concern that a proportion of the veterinarians’ practice did not adhere to this stewardship principle, and some of the respondents did not rely on culture and sensitivity tests and drug guidelines. These findings contradict the observations of other researchers [44,45,46], who recorded that majority of the veterinarians were influenced by the antibiotic susceptibility test (AST) in the selection of antimicrobials. Ordering ASTs also depends on the availability of the facilities and on cost-effectiveness [46,47]. Veterinary diagnostic facilities are limited in Bangladesh, including ASTs. Even an aware veterinarian often cannot use ASTs, even if there is treatment failure, because of the high cost. Lack of facilities for ASTs and the higher cost associated with existing AST facilities in the country, leading to their lack of affordability for farmers, might be the factors leading to not ordering ASTs in the selection of antimicrobials, as described in the literature [48]. These factors may further lead to inappropriate use of antimicrobials in terms of using additional antimicrobials in a single prescription, or overdosing of antimicrobials [49,50]. However, veterinarians overcome these limitations by using their own experience, sharing the experience of senior colleagues, previous training, and gathering information from different sources, and this was also supported by the findings of other studies [42,44].

Although veterinarians had relatively good KAP on AMR and AMU, factors related to KAP have demonstrated important gaps in the KAP, which were also reported by a good proportion of vets in the current study. The findings of this study were in line with those of other researchers who showed that the veterinarians had good knowledge of AMU and AMR issues compared to other stakeholders such as poultry feed and drug sellers [32,33,37,42,43]. Our study demonstrated that a higher proportion of veterinarians had poor-to-medium knowledge on classes and generations of antibiotics and on selecting the correct antimicrobials, including their doses and routes of administration. More than half of the respondents had a medium level of knowledge on interpreting laboratory results, and a majority of them were unaware of the reserve group of antibiotics. This could be due to lack of awareness, poorly updated knowledge, level of education and years of experience, availability of treatment or drug guidelines, and implementation of these by the regulatory bodies [42,43,45]. In our study, although most veterinarians agreed that there is a threat of AMR in livestock and poultry production sectors, and with statements on the selection of correct antimicrobials and their sources and irrational uses of antibiotics, a proportion of the respondents had inappropriate knowledge on these topics. We found that veterinarians’ age, level of education, years of experience, and previous training on AMU and AMR could be the possible reasons, and these findings are aligned with those of other studies [43,44,45]. Our study indicates that more educated, experienced, and trained veterinarians have improved KAP regarding AMU and AMR information. However, a proportion of veterinarians were not adequately cautious about the right sources of information. They particularly depended on prior training, their own experience and experience shared by colleagues. The majority did not follow the national (NAP AMR) and international (WHO) guidelines on AMR. Furthermore, inappropriate administration and overdosing of antibiotics by the unaware veterinarians could be an important factor contributing to the development of AMR observed by previous studies [10,43,51]. A low level of KAP on AMU and AMR can ultimately lead to over-prescription, increasing the risk of misuse of antibiotics in animals [33,43].

The study further showed that veterinarians’ practices in providing regular advice to the farmers on withdrawal of antimicrobials, recording of the antimicrobials they use, and completion of the entire course of an antimicrobial. However, a good proportion of the veterinarians followed poor practices. Many of them did not use ASTs, there were frequent uses of more than one antibiotic in a single prescription, antimicrobials were prescribed based on the demands of the farmers and without clinical monitoring of the animals, and higher doses of antibiotics were prescribed for rapid recovery. Prescribing of antimicrobials based on the client demand without clinical observations of the animals was not avoided by a proportion of the Bangladeshi veterinarians, and this is considered a bad practice which may aggravate the AMR problem [43,45]. In Bangladesh, the veterinary services are unable to reach a significant proportion of farmers due to the lack of veterinarians. Often farmers in remote areas find it hard to transport sick animals to veterinary hospitals due to high transport costs, poor transport facilities, and the poor health condition of the animal. As a result, veterinarians must give antimicrobials without visiting the farms, whether willingly or unwillingly, considering the scarcity of services available in those areas. Sometimes, the fear of losing clients and the attempt to avoid treatment by para-vets or quacks are other reasons for such prescription behavior, as also observed in other countries [47]. It is also noted that hard-to-reach areas lack a sufficient number of veterinarians, resulting in veterinary services being covered by other unprofessional practitioners (such as para-vets or quacks, or even feed and drug dealers) who indiscriminately store, distribute, and use antimicrobials for the treatment of animals, resulting in a worsening of the AMR situation in the country [32,33,43].

Veterinarians’ roles are crucial in changing the behavior of almost all the stakeholders involved in veterinary practices [32,37,42,43]. Those without the level of KAP of veterinarians have a negative impact in disseminating inappropriate advice to stakeholders regarding changing their behavior on AMU and AMR and implementing the AMR policy of the country. This suggests a need for more in-depth and nationwide training and awareness programs for animal health workers, including veterinarians, to curb the development of AMR in the country. Some research also showed that one-on-one meetings, with the dissemination of resource materials, were positively associated with changing farmers’ behavior regarding AMR [42]. In alignment with other studies, [42,43,49], the current investigation showed that the level of education, experience, and training on AMU and AMR, not only for veterinarians but also for other animal health workers, could be crucial in altering the prescribing behavior of farmers.

## 5. Conclusions

This study demonstrated that a number of clinical etiological factors, such as the use of susceptibility tests or the inability to interpret such reports, influenced the selection of appropriate antimicrobials. Other factors such as the economic condition of the farmer, locally available antimicrobials, and drug instructions had a high-to-moderate influence when selecting an antimicrobial. The study further revealed that veterinarians’ socio-economic and demographic characteristics such as age, level of education, years of experience, and previous training, were the key factors in their knowledge, attitudes, and practices regarding AMU and AMR. Considering the results of our study, the indiscriminate use of AMU in Bangladesh can be alleviated by the inclusion of AMR in the veterinary curricula, continuous education, and awareness campaigns, with more training for practicing veterinarians and other stakeholders, easy access and dissemination, and implementation of AMR resources, for example, preparing a database of antimicrobial use and increasing the provision of veterinary services in hard-to-reach areas of the country.

## Figures and Tables

**Figure 1 antibiotics-11-00080-f001:**
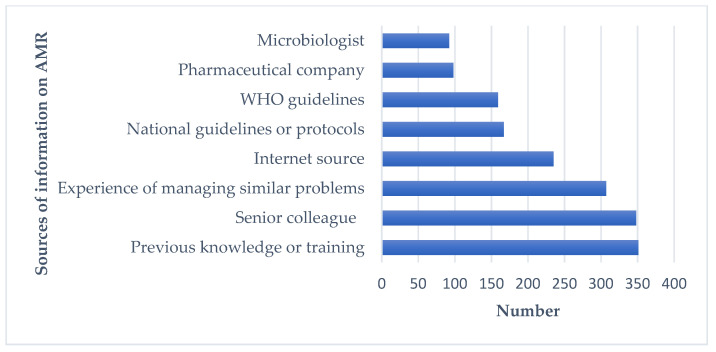
Sources of information on AMR for participants.

**Figure 2 antibiotics-11-00080-f002:**
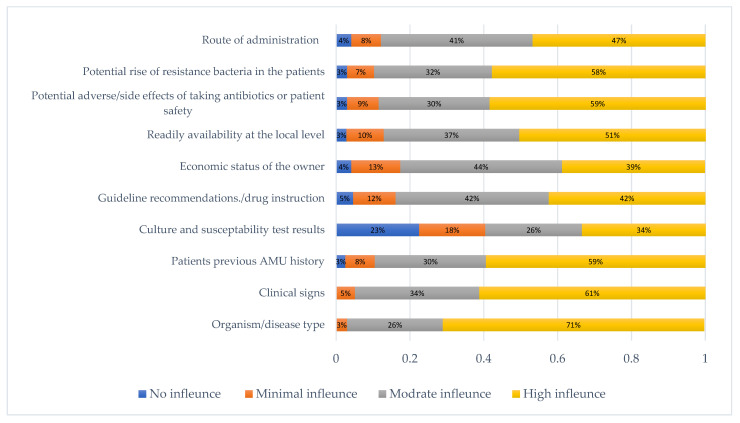
Clinical etiological and other factors influencing the selection of antimicrobial prescription or administration by the veterinarians.

**Table 1 antibiotics-11-00080-t001:** Demographic characteristics of study participants.

Variables		*n* (%)
Respondent’s gender	Female	73 (16.7)
	Male	363 (83.3)
Age (years)	18–25	159 (36.5)
	26–30	160 (36.7)
	31–35	55 (12.6)
	36–40	41 (9.4)
	41 or more	21 (4.82)
Level of education	DVM	284 (65.14)
	Master’s/post-graduate	152 (34.86)
Experience (years)	Intern	168 (38.5)
	Up to 3	134 (30.7)
	4–6	50 (11.5)
	7 or more	84 (19.3)
Current workplace	Private	136 (31.2)
	Government hospital	235 (53.9)
	Medicine/feed company	65 (14.9)
Training on antimicrobial use	Non-trained	246 (56.4)
	Trained	190 (43.6)

**Table 2 antibiotics-11-00080-t002:** Knowledge of veterinarians on AMU and AMR.

Items	Not at All N (%)	Poor N (%)	Medium N (%)	Good N (%)
Knowledge of different classes and generations of antibiotics	-	6 (1.4)	155 (35.6)	275 (63.1)
Knowledge on interpreting microbiological/ laboratory results	3 (0.7)	57 (13.1)	217 (49.8)	159 (36.5)
Knowledge on choosing the correct antimicrobial	1 (0.2)	14 (3.2)	194 (44.5)	227 (52.1)
Knowledge on choosing the correct dose/dosage of antimicrobials	2 (0.5)	8 (1.8)	144 (33.0)	282 (64.7)
Knowledge on choosing routes of antimicrobial administration (oral vs. intravenous vs. topical)	-	7 (1.6)	102 (23.4)	327 (75.0)
Knowledge on using a combination of antimicrobials if appropriate	3 (0.7)	33 (7.6)	243 (55.7)	157 (36.0)
Knowledge on planning the duration of the specific antimicrobial treatment	3 (0.7)	21 (4.8)	189 (43.4)	223 (51.2)
Knowledge on modifying/stopping antimicrobial treatments if required	3 (0.7)	35 (8.0)	210 (48.2)	188 (43.1)
Knowledge about reserve group of antimicrobials	9 (2.1)	61 (14.0)	181 (41.5)	185 (42.4)
Knowledge of critically important list of antimicrobials specified by World Health Organization (WHO)	19 (4.4)	76 (17.4)	178 (40.8)	163 (37.4)
Knowledge of National Action Plan for Antimicrobial Resistance (NAP AMR)	15 (3.4)	78 (17.9)	211 (48.4)	132 (30.3)
Knowledge on the mechanism and causes of AMR	9 (2.1)	37 (8.5)	127 (29.1)	263 (60.3)

**Table 3 antibiotics-11-00080-t003:** Attitudes of veterinarians on AMU and AMR.

Items	Strongly Disagree N (%)	Disagree N (%)	Agree N (%)	Strongly Agree N (%)
Antimicrobial resistance is a big threat for livestock and Poultry production	2 (0.5)	3 (0.7)	47 (10.8)	384 (88.1)
A single course of antibiotics can cause antimicrobial resistance	12 (2.8)	90 (20.6)	170 (39.0)	164 (37.6)
Irrational antibiotic use in animals leads to antibiotic resistance in humans	2 (0.5)	17 (3.9)	152 (34.9)	265 (60.8)
Antimicrobial resistance is a natural as well as anthropogenic phenomenon	15 (3.4)	99 (22.7)	216 (49.5)	106 (24.3)
Antimicrobial resistance will become a greater clinical problem in the future than it is today	-	8 (1.8)	77 (17.7)	351 (80.5)
In recent years I have become more aware of the impacts of antimicrobial resistance	2 (0.5)	11 (2.5)	149 (34.2)	274 (62.8)
I find it hard to select the correct antimicrobial	7 (1.6)	80 (18.4)	229 (52.5)	120 (27.5)
I have enough sources of information about antimicrobials and their uses	6 (1.4)	81 (18.6)	230 (52.8)	119 (27.3)
New antimicrobials will be developed that will keep up with the problem of antimicrobial resistance	15 (3.4)	85 (19.5)	226 (51.8)	110 (25.2)
Restricting “priority antibiotics” for human use only	18 (4.1)	66 (15.1)	157 (36.0)	195 (44.7)

**Table 4 antibiotics-11-00080-t004:** Practices of veterinarians on AMU and AMR.

Items	Never N (%)	Rarely N (%)	Frequently N (%)	Regularly N (%)
How often do you give advice about the withdrawal period of antimicrobials?	8 (1.8)	75 (17.2)	116 (26.6)	237 (54.4)
How often do you give advice to the farmers to keep records of antimicrobials?	16 (3.7)	61 (14.0)	153 (35.1)	206 (47.3)
How often do you advise the farmer on administering antimicrobials through telephone conversations?	51 (11.7)	172 (39.5)	127 (29.1)	86 (19.7)
How often do you use antibiotics for prophylaxis?	57 (13.1)	154 (35.3)	162 (37.2)	63 (14.5)
How often do you use bacterial culture and susceptibility testing to select the most appropriate antibiotics for your treatment?	115 (26.4)	176 (40.4)	99 (22.71)	46 (10.6)
How often do you prescribe more than one antimicrobial in a single prescription?	64 (14.7)	200 (45.8)	127 (29.1)	45 (10.3)
How often do you advise the farmer about completing the full course of antimicrobials that you prescribed?	2 (0.5)	14 (3.2)	68 (15.6)	352 (80.7)
How often do you use antimicrobials due to the demand of farmers in a situation which does not require their use?	146 (33.5)	134 (30.7)	93 (21.3)	63 (14.5)
How often do you write prescriptions for antimicrobials to farmers who come to you without their animals?	82 (18.8)	163 (37.4)	136 (31.2)	55 (12.6)
How often do you use a higher dose of antimicrobials for rapid recovery of your patient?	71 (16.3)	196 (45.0)	120 (27.5)	49 (11.2)
How often do you use different alternatives of antimicrobials?	12 (2.8)	137 (31.4)	217 (49.8)	70 (16.1)
How often do you advise farmers about proper vaccination to reduce the use of antimicrobials?	5 (1.2)	20 (4.6)	81 (18.6)	330 (75.7)

**Table 5 antibiotics-11-00080-t005:** Test of statistical significance of variation in the respondents’ knowledge on AMU and AMR by their characteristics.

Variables		Knowledge				Attitudes				Practices			
		Incorrect N (%)	Moderate N (%)	Correct N (%)	*p*	Unfavorable N (%)	Moderate N (%)	Favorable N (%)	*p*	Bad: N (%)	Moderate N (%)	Good: N (%)	*p*
**Gender**	Female	7 (9.6)	35 (48.0)	31 (42.5)	0.502	6 (8.2)	37 (50.7)	30 (41.1)	0.472	4 (5.5)	48 (65.8)	21 (28.8)	0.016
	Male	28 (7.7)	154 (42.4)	181 (42.4)		33 (9.1)	208 (57.3)	122 (33.6)		70 (19.3)	206 (56.8)	87 (24.0)	
**Age (years)**	18–25	12 (7.6)	84 (52.8)	63 (39.6)	0.003	11 (6.9)	92 (57.9)	56 (35.2)	0.720	24 (15.1)	104 (65.4)	31 (19.5)	0.198
	26–30	14 (8.8)	70 (43.8)	76 (47.5)		15 (9.4)	90 (56.3)	55 (34.4)		30 (18.8)	91 (56.9)	39 (24.4)	
	31–35	3 (5.5)	22 (40.0)	30 (54.6)		6 (10.9)	28 (50.9)	21 (38.2)		11 (20.0)	28 (50.9)	16 (29.1)	
	36–40	5 (12.2))	11 (26.8)	25 (61.0)		5 (12.2)	26 (63.4)	10 (24.4)		8 (19.5)	19 (46.3)	14 (34.2)	
	41 or more	1 (4.8)	2 (9.5)	18 (85.7)		2 (9.5)	9 (42.9)	10 (47.6)		1 (4.8)	12 (57.1)	8 (38.1)	
**Level of education**	Undergraduate	25 (8.8)	134 (47.2)	125 (44.0)	0.031	19 (6.7)	167 (58.8)	98 (34.5)	0.059	49 (17.3)	176 (62.0)	59 (20.8)	0.027
	Master’s/post-graduate	10 (6.6)	55 (36.2)	152 (57.2)		20 (13.2)	78 ((51.3)	54 (35.5)		25 (16.5)	78 (51.3)	49 (32.2)	
**Years of experience**	Intern	14 (8.3)	88 (52.4)	66 (39.3)	0.018	13 (7.7)	94 (56.0)	61 (36.3)	0.526	27 (16.1)	110 (65.5)	31 (18.5)	0.124
	Up to 3	10 (7.5)	59 (44.0)	65 (48.5)		13 (9.7)	77 (57.5)	44 (32.8)		27 (20.2)	73 (54.5)	34 (25.4)	
	4–6	5 (10.0)	17 (34.0)	28 (56.0)		6 (12.0)	32 (64.0)	12 (24.0)		8 (16.0)	28 (56.0)	14 (28.0)	
	7 or more	6 (7.1)	25 (29.8)	53 (63.1)		7 (8.3)	42 (50.0)	35 (41.7)		12 (14.3)	43 (51.2)	29 (34.5)	
**Current workplace**	Private practice	11 (8.1)	60 (44.1)	65 (47.8)	0.562	15 (11.0)	72 (52.9)	49 (36.0)	0.196	29 (21.3)	68 (50.0)	39 (28.7)	0.170
	Government hospital	17 (7.2)	97 (41.3)	121 (51.5)		21 (8.9)	128 (54.5)	86 (36.6)		37 (15.7)	146 (62.1)	52 (22.1)	
	Medicine/feed company	7 (10.8)	32 (49.2)	65 (40.0)		3 (4.6)	45 (69.2)	17 (26.2)		8 (12.3)	40 (61.5)	17 (26.2)	
**Training on AMU and AMR**	No training	23 (9.4)	126 (51.2)	97 (39.4)	0.000	27 (11.0)	138 (56.1)	81 (32.9)	0.201	39 (15.9)	155 (63.0)	52 (21.1)	0.060
	Received training	12 (6.3)	63 (33.2)	115 (60.5)		12 (6.3)	107 (56.3)	71 (37.8)		35 (18.4)	99 (52.1)	56 (29.5)	

**Table 6 antibiotics-11-00080-t006:** Logistic regression analysis of the factors associated with respondents’ knowledge, attitudes, and practices on AMU and AMR.

Variables		Knowledge	Attitudes	Practices
		OR, 95%CI, *p*	OR, 95%CI, *p*	OR, 95%CI, *p*
Gender	Female	Ref	Ref	Ref
	Male	1.38, 0.79–2.38, 0.257	1.10, 0.63–1.93, 0.729	0.59, 0.32–1.06, 0.077
Age (years)	18–25	Ref	Ref	Ref
	26–30	0.92, 0.46–1.84, 0.814	1.20, 0.60–2.45, 0.608	0.66, 0.33–1.34, 0.252
	31–35	0.75, 0.25–2.30,0.620	0.44, 0.14–1.35, 0.152	1.38, 0.45–4.26, 0.570
	36–40	0.81, 0.20–3.20, 0.760	0.24, 0.06–0.97, 0.043	0.79, 0.20–3.15, 0.733
	41 or more	2.71, 0.38–19.3, 0.319	0.29, 0.06–1.54, 0.147	0.81, 0.16–4.18, 0.799
Level of education	Undergraduate	Ref	Ref	Ref
	Master’s/post-graduate	1.23, 0.71–2.12, 0.465	1.16, 0.67–2.00, 0.598	1.33, 0.78–2.27, 0.295
Experience (years)	Intern	Ref	Ref	Ref
	Up to 3	1.11, 0.52–2.38, 0.779	1.02, 0.47–2.21, 0.954	0.95, 0.44–2.04, 0.896
	4–6	2.03, 0.70–5.89, 0.193	1.69, 0.57–4.99, 0.345	0.79, 0.28–2.19, 0.645
	7 or more	1.49, 0.41–5.50, 0.547	3.63, 0.95–13.95, 0.060	1.53, 0.40–5.87, 0.534
Current workplace	Medicine/feed company	Ref	Ref	Ref
	Private practice	1.48, 0.77–2.85, 0.244	0.96, 0.48–1.89, 0.899	0.83, 0.42–1.65, 0.599
	Government hospital	2.09, 1.06–4.10, 0.032	1.15, 0.58–2.28, 0.698	0.60, 0.30–1.19, 0.154
Training	No training	Ref	Ref	Ref
	Trained	1.92, 1.23–2.97, 0.004	2.09, 1.35–3.25, 0.001	0.76, 0.50–1.16, 0.024

## Data Availability

The data presented in this study are available in the article and Appendix A.

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
