# Peer review of "Knowledge, Attitudes, and Common Practices of Livestock and Poultry Veterinary Practitioners Regarding the AMU and AMR in Bangladesh"

_antibiotics, 2022, doi:10.3390/antibiotics11010080_

Round 1

Reviewer 1 Report

Brief summary 

This article aims to analyze the factors influencing the prescription of antibiotics in Bangladesh in veterinary medicine. The authors constructed a questionnaire that was sent to veterinarians. The questionnaires consisted of several questions regarding the level of knowledge of veterinarians about AMR and AMU, clinical etiological factors influencing antimicrobial prescription, attitude factors and practice factors. 436 responses were obtained and analyzed. Authors conclude that drivers of antimicrobial prescription are clinical etiological factors, economic factors and knowledge. They suggest that teaching and communication of AMR might be improved.

General concept comments

The objective of the study is interesting and corresponds to the scope of the journal. There are still some minor errors (spelling, typing errors). I think that results might be more analyzed and discussed (see specific comments).

Specific comments 

the title is very similar to other papers already published

Line 34 : explicit AMR as it is the first time mentioned

Line 36 : extra space between words between "the" and "inappropriate"

Line 60 : please explain what is "animal industries" is it industrial farming ?

line 74 : extra space between words between "that" and "bangladesh"

line 85 : authors talk about a National action plan without any reference. Please add it and explain the objective of this action plan

line 128 : section is repeated, please re written the sentence

line 130 : I don't understand "there negative and seven positive question ? please explain.

Table 1 :  in the category "level of education" the total of Intern + Masters don't match with ne number of response 436. Geographical data were not explored ?

figure 1 : can they choose several proposals for multiple choice question and then could you classify the choice ?

figure 2 : there are spelling mistake in the legend

Table 2 : why in the first line of the table to total don't reach 436 ? a parentheses is missing on line 2 of the table

Line 277 : I don't understand this sentence. I have understand that CS is not really done so why author say "highest number"

Table 4 : I do not see the explanation of the asterisk. extra space between words between "different" and "alternatives"

line 307 :  extra space between words between "well" and "while"

line 320 :  extra space between words between "factors" and "associated"

line 322 : I'm not agree with the comment, disease, clinical signs and symptoms are not discussed in this paper

line 350 : there is a mistake "by the a mentionable vets"

line 441 : mistake on word acknowledge. same remarks on line 413 "available"

Other questions

Does authors try to associate some response. For example, in line 253, is there any correlation with people who answer having good knowledge on antimicrobials.

In the same idea, is there any correlation with people who answer having good knowledge and study level ?

The majority are in government hospital, these people are correlated with good knowledge ? is there any correlation with the use of CS ...

Around 40% of respondents consider that there is no link between use and AMR. It seems to be very high, is it the same persons who respond to have sufficient knowledge.

line 256 concerning persons who answer to have difficulties to choose antibiotics (80%), how many of them think they have sufficient information ? 

It will be interesting to compare some results with pets studies.

Is cost of antibiotics a driver of antibiotics choice?

Do you have a geographical repartition of responses?

Are all questionnaires complete ?

I think etiological clinical is not a good term because there are no clinical discussion of antimicrobial use

Epidemiological data are available to choice antibiotics and help vets to choose instead of having CS?

Proposition of paper to be taking in account to complete discussion

Chapot L, Sarker MS, Begum R, Hossain D, Akter R, Hasan MM, Bupasha ZB, Bayzid M, Salauddin M, Parvej MS, Uddin AM, Hoque F, Chowdhury J, Ullah MN, Rahman MK, Siddiky NA, Fournié G, Samad MA. Knowledge, Attitudes and Practices Regarding Antibiotic Use and Resistance among Veterinary Students in Bangladesh. Antibiotics (Basel). 2021 Mar 22;10(3):332. doi: 10.3390/antibiotics10030332. PMID: 33809932; PMCID: PMC8004205.

Odoi A, Samuels R, Carter CN, Smith J. Antibiotic prescription practices and opinions regarding antimicrobial resistance among veterinarians in Kentucky, USA. PLoS One. 2021 Apr 15;16(4):e0249653. doi: 10.1371/journal.pone.0249653. PMID: 33857198; PMCID: PMC8049335.

Tompson AC, Chandler CIR, Mateus ALP, O'Neill DG, Chang YM, Brodbelt DC. What drives antimicrobial prescribing for companion animals? A mixed-methods study of UK veterinary clinics. Prev Vet Med. 2020 Oct;183:105117. doi: 10.1016/j.prevetmed.2020.105117. Epub 2020 Aug 5. PMID: 32890918.

Author Response

Brief summary 

This article aims to analyze the factors influencing the prescription of antibiotics in Bangladesh in veterinary medicine. The authors constructed a questionnaire that was sent to veterinarians. The questionnaires consisted of several questions regarding the level of knowledge of veterinarians about AMR and AMU, clinical etiological factors influencing antimicrobial prescription, attitude factors and practice factors. 436 responses were obtained and analyzed. Authors conclude that drivers of antimicrobial prescription are clinical etiological factors, economic factors and knowledge. They suggest that teaching and communication of AMR might be improved.

Response: Thank you so much for your close review on our paper. We appreciate your time and effort you provided to make comments, raise questions and make suggestions.

General concept comments

The objective of the study is interesting and corresponds to the scope of the journal. There are still some minor errors (spelling, typing errors). I think that results might be more analyzed and discussed (see specific comments).

Response: Thank you so much for your appreciation. We have now made changes, modifications and revisions based on your comments, suggestions and questions.  

Specific comments 

The title is very similar to other papers already published

Response: Thank you so much for your comment. A number of papers have been published in the literature. However, to our knowledge, this is the first study in the country that included registered veterinarians who are practising in both poultry and livestock sector.

Line 34: explicit AMR as it is the first time mentioned

Response: Thanks so much for pointing out this. It is now spelled out in the first appearance. Please check line number 35 in the revised version.

Line 36 : extra space between words between "the" and "inappropriate"

Response: Thanks so much for pointing out this. Made necessary corrections.

Line 60: please explain what is "animal industries" is it industrial farming?

Response: Thanks so much for your comment. Yes, we meant industrial farming. Made necessary correction. Please check line number 61-62 in the revised version.

line 74 : extra space between words between "that" and "bangladesh"

Response: Thanks so much for pointing out this. Made necessary corrections.

line 85: authors talk about a National action plan without any reference. Please add it and explain the objective of this action plan

Response: Thanks so much for pointing out this. The reference has been added. Please see reference number 34 in the revised version.

line 128: section is repeated, please re written the sentence

Response: Thanks so much for pointing out this. Made necessary corrections in the revised version. Please check line number 132-134 in the revised version.

line 130: I don't understand "there negative and seven positive question? please explain.

Response: Thanks so much for your comment. While developing the questions, we added a few questions, which are not favorable in terms of AMR and AMU. Similarly, we added some positive questions. These questions were developed based on literature reading and reviewing.

Table 1:  in the category "level of education" the total of Intern + Masters don't match with ne number of responses 436. Geographical data were not explored?

Response: Thanks so much for pointing out this. It was a typo. Made necessary corrections in the table 1 of the revised version.

Regarding the geographical data, we did not collect this information since we were unable to collect data physically.

figure 1 : can they choose several proposals for multiple choice question and then could you classify the choice ?

Response: Thanks so much for your question. To understand the commonly used sources of information of the vets, we asked a question with multiple responses. The figure was built based on those responses.

Figure 2 : there are spelling mistake in the legend

Response: Thanks so much for pointing out this. It was a typo. Made necessary corrections in the revised version. Please see the revised version.  

Table 2 : why in the first line of the table to total don't reach 436 ? a parentheses is missing on line 2 of the table

Response: Thanks so much for pointing out this. It was a typo and it occurred while transferring the results into the draft. Made necessary corrections in the revised version.

Line 277 : I don't understand this sentence. I have understand that CS is not really done so why author say "highest number"

Response: Thanks so much for pointing out this. We have now made necessary changes in the revised version. Please check line number 289 in the revised version.

Table 4 : I do not see the explanation of the asterisk. extra space between words between "different" and "alternatives"

Response: Thanks so much for pointing out this. It was a typo. We have now deleted the asterisk from the revised version.

line 307 :  extra space between words between "well" and "while"

Response: Thanks so much for pointing out this. We have now made necessary changes in the revised version.

line 320 :  extra space between words between "factors" and "associated"

Response: Thanks so much for pointing out this. We have now made necessary changes in the revised version.

line 322 : I'm not agree with the comment, disease, clinical signs and symptoms are not discussed in this paper

Response: Thanks so much for comment. This comment was made based on the findings of Figure 2 where it shows that disease or organisms, clinical signs and symptoms, history, and potential side effects of the antimicrobials were some of the key factors that influence prescribing antimicrobials. However, we have now made changes on that comment. Please check line number 334 to 339 in the revised version.

line 350 : there is a mistake "by the a mentionable vets"

Response: Thanks so much for pointing out this. We have now made necessary changes in the revised version. Please check line number 364 in the revised version.

line 441 : mistake on word acknowledge. same remarks on line 413 "available"

Response: Thanks so much for pointing out this. We have now made necessary changes in the revised version.

Other questions

Does authors try to associate some response. For example, in line 253, is there any correlation with people who answer having good knowledge on antimicrobials.

Response: Thanks so much for your comment. In table 3, we intended to understand the perception of veterinarians by providing a few positive and negative statements on antimicrobial resistance. Therefore, we did not correlate them with the knowledge of antimicrobials. 

In the same idea, is there any correlation with people who answer having good knowledge and study level?

Response: Thanks so much for your comment. In table 5, we presented association between the level of knowledge and the level of study. We categorized the responses regarding knowledge questions were coded into three-point index- Incorrect, moderate, and correct.  

The majority are in government hospital, these people are correlated with good knowledge? Is there any correlation with the use of CS ...

Response: Thanks so much for your comment. In general, the scope of culture and susceptibility (CS) test is very limited in veterinary sector in Bangladesh. Government veterinary hospitals also have limitations of such facilities, so we believe correlation analysis would have limited value to the current analysis on good knowledge of participants from government hospital. The table 5 shows the results. However, the multinomial regression analysis shows the vets who were working in the government facilities, had 2.09 times more likely to have better knowledge compared to those who were working in the Medicine/feed company. However, in regards to attitudes and practices, there were no significant association. Meaning, facilities in the government vet hospitals did not have any influence on attitude and practices.

Around 40% of respondents consider that there is no link between use and AMR. It seems to be very high, is it the same persons who respond to have sufficient knowledge.

Response: Thanks so much for your comment. In table 5, we presented association between the level of knowledge and the level of study. We categorized the responses regarding knowledge questions were coded into three-point index- Incorrect, moderate, and correct. 

line 256 concerning persons who answer to have difficulties to choose antibiotics (80%), how many of them think they have sufficient information? 

Response: Thanks so much for your comment. For our understandings, it is a bit difficult to figure that number since, the question on sources of information was multiple response type and on the other hand, the attitude question was one response question.    

It will be interesting to compare some results with pets’ studies.

Response: Thanks so much for your comment. The aim of our study was focused to understand KAP of those vets who were engaged in livestock and poultry sector. Therefore, it is out of aim to compare the results with pet vets.     

Is cost of antibiotics a driver of antibiotics choice?

Response: Thanks so much for your comment. In general, the cost of antibiotics is low compared to other countries. However, we did not include this as a driver of choosing antibiotics.     

Do you have a geographical repartition of responses?

Response: Thanks so much for your comment. We did not include this in our study as the study was conducted using online questionnaire and we were not able to collect real time location of the respondents.   

Are all questionnaires complete?

Response: Thanks so much for your comment. We only considered complete responses in this analysis. 

I think etiological clinical is not a good term because there are no clinical discussion of antimicrobial use

Response: Thanks so much for pointing out this. We have changed this term.

Epidemiological data are available to choose antibiotics and help vets to choose instead of having CS?

Response: Thanks for your comment. In Bangladesh there was no surveillance system for AMR, therefore, epidemiological data were not available. 

Proposition of paper to be taking in account to complete discussion

Chapot L, Sarker MS, Begum R, Hossain D, Akter R, Hasan MM, Bupasha ZB, Bayzid M, Salauddin M, Parvej MS, Uddin AM, Hoque F, Chowdhury J, Ullah MN, Rahman MK, Siddiky NA, Fournié G, Samad MA. Knowledge, Attitudes and Practices Regarding Antibiotic Use and Resistance among Veterinary Students in Bangladesh. Antibiotics (Basel). 2021 Mar 22;10(3):332. doi: 10.3390/antibiotics10030332. PMID: 33809932; PMCID: PMC8004205.

Odoi A, Samuels R, Carter CN, Smith J. Antibiotic prescription practices and opinions regarding antimicrobial resistance among veterinarians in Kentucky, USA. PLoS One. 2021 Apr 15;16(4):e0249653. doi: 10.1371/journal.pone.0249653. PMID: 33857198; PMCID: PMC8049335.

Tompson AC, Chandler CIR, Mateus ALP, O'Neill DG, Chang YM, Brodbelt DC. What drives antimicrobial prescribing for companion animals? A mixed-methods study of UK veterinary clinics. Prev Vet Med. 2020 Oct;183:105117. doi: 10.1016/j.prevetmed.2020.105117. Epub 2020 Aug 5. PMID: 32890918.

Response: Thanks so much for suggesting these articles. We have citated these papers. Please see citation numbers 48, 49 and 50 in the revised version.

Reviewer 2 Report

This is an interesting manuscript regarding KAP in veterinarians in your country. Nevertheless, some points must be improved.

The abstract must be revised when the main text was updated.

Line 75. “Animal-source protein” is meat and milk, that is, pigs and ruminants?

Line 82. “Evolution of different ..”? Please, explain.

Lines 99-102. Is this phrase related to your country?

Line 119. Although I suppose the questionnaire was in a local language, a copy available upon request should be offered.

Line 145. How many vets where in this list?

Line 148 and table 1. “”interns”? Please, explain.

Line 150. It is not clear the distinction between poultry and animal. Please, clarify.

Line 157. The number of responders is a result.

Lines 188-193. These categorizations and the “three-point index” must be explained with more detail.

Table 1 (and text). According to line 112 “target participants were livestock and poultry veterinarians”. This distribution must be added to the result section. Livestock is only ruminants?

Table 1. Age and experience. If these variables were collected numerically, please summarize the data as a quantitative variable (mean, median, percentiles, etc.)

Table 1. “Training on antimicrobial use”. Do you have more specific information about the training?

Lines 209 and 213. Please explain how this information was retrieved. It was an open question or a multiple closed-answers question?

Lines 281-225. Please explain how this information was retrieved. It was an open question or a multiple closed-answers question?

Table 2 (and lines 239, 358). What is “reserve group” of antimicrobials?

Table 2. Some questions are very broad (first, last) so the results are of low interest.

Points 3.2 to 3.6 and table 2 to 5. Separate results for livestock and poultry veterinarians should be presented or, at least, differences or not between them must be stated.

Table 6. It is not clear how the logistic regression analysis is performed. Please, explain.

Line 323. I do not see any question regarding signs. In addition, animals have not symptoms.

Line 325. “Better” than..?

Lines 329-330. I do not see this data on the results section. Please, add before discussing.

Line 332. “Infield”?

Lines 342-343. “Confounding factors” are not been analysed in this work. Please, delete this sentence.

Lines 349-350. No sense.

Line 353. Please explain what others stakeholders.

Lines 354 and 379.   - “Good” is not adequate. A proportion could be high, medium, low, etc.

Lines 366-368. I do not agree with these sentences regarding your results.

Lines 376-378. No sense.

Line 386. “Shortage”?

Lines 389-390. Is this prescription without visiting the farms?

Line 407. “Other animal health workers”? This manuscript only focus on vets.

Line 410. “Aetilogial factors”? In addition, not use of AST and inability for interpretation should be separated.

Author Response

This is an interesting manuscript regarding KAP in veterinarians in your country. Nevertheless, some points must be improved.

Response: Thank you so much for your appreciation. We have now made necessary changes and revisions based on the comments and suggestions. We really appreciate your time and efforts you gave to our manuscript.

The abstract must be revised when the main text was updated.

Response: Thanks so much for your suggestion. We have updated the abstract based on the modifications we made in the main text.

Line 75. “Animal-source protein” is meat and milk, that is, pigs and ruminants?

Response: Thanks so much for pointing out this. We have now specified with chicken, eggs, milk and meat. Please check line number 77-78 in the revised version.

Line 82. “Evolution of different ..”? Please, explain.

Response: Thanks for your comment. We meant develop of AMR pathogens with zoonotic importance. We have replaced ‘evolution’ with ‘development’.  Please check line number 85-86 in the revised version.

Response:

Lines 99-102. Is this phrase related to your country?

Response: Thanks so much for pointing out this. We have now specified Bangladesh. Please check line number 106 in the revised version.

Line 119. Although I suppose the questionnaire was in a local language, a copy available upon request should be offered.

Response: Thanks so much for your suggestion. The primary version of the questionnaire was developed in English. The questionnaire will be available upon request to corresponding author.

Line 145. How many vets where in this list?

Response: Thanks so much for your question. The number of veterinarians has been added in the parentheses. Please check line number 153 in the revised version.

Line 148 and table 1. “”interns”? Please, explain.

Response: Thanks so much for your question. It was a typo. We have now made corrections in table 1.

Line 150. It is not clear the distinction between poultry and animal. Please, clarify.

Response: Thanks so much for your question. It was a typo. We have now updated this with livestock. Please check line number 157 in the revised version. 

Line 157. The number of responders is a result.

Response: Thanks so much for your comment. We agree this is a result. However, to give the better idea on how many vets were included in the study in regards to the number of vets were approached, we believe this is more suited to mention in the methodology section, specifically in the sampling sub-section.  

Lines 188-193. These categorizations and the “three-point index” must be explained with more detail. Please see line number 198-202 in the revised version.

Response: Thanks so much for your comment. We have now explained this in the revised version.

Table 1 (and text). According to line 112 “target participants were livestock and poultry veterinarians”. This distribution must be added to the result section. Livestock is only ruminants?

Response: Thanks so much for your suggestions. In Bangladesh, the majority of the vets practice in both poultry and livestock sector simultaneously. It was also verified while collecting the list of vets from Bangladesh veterinary council. However, a very few vets are working either in poultry or livestock sector specifically. As it is found that almost all vets are working in both livestock and poultry sectors, as a result, it is not possible for us to show vets’ nature wise disaggregated analysis. But we definitely agree with your suggestion.

Table 1. Age and experience. If these variables were collected numerically, please summarize the data as a quantitative variable (mean, median, percentiles, etc.)

Response: Thanks so much for your suggestion. We did not collect specific numeric data on age and years of experience. We collected these data in age and years of experience in group.

Table 1. “Training on antimicrobial use”. Do you have more specific information about the training?

Response: Thanks so much for your question. By this question, we wanted to understand if the sampled vets received any types of training related to AMU in general. We did not have any follow-up questions regarding the specific aspects of training. However, this close-ended question enabled us to understand the association between KAP in regards to those who received any training and those who did not receive any. 

Lines 209 and 213. Please explain how this information was retrieved. It was an open question or a multiple closed-answers question?

Response: Thanks so much for your comment. We collected this information using a close ended multiple options question.

Table 2 (and lines 239, 358). What is “reserve group” of antimicrobials?

Response: Thanks so much for your question. We followed World Health Organization’s recommendation on the reserved group of antimicrobials. WHO defines reserved groups as “This group includes antibiotics that should be treated as ‘last-resort’ options, or tailored to highly specific patients and settings, and when other alternatives would be inadequate or have already failed (e.g., serious life-threatening infections due to multi-drug resistant bacteria). These medicines could be protected and prioritized as key targets of high-intensity national and international stewardship programs involving monitoring and utilization reporting, to preserve their effectiveness. Eight antibiotics or antibiotic classes were identified for this group.” For details, please see at https://www.who.int/medicines/publications/essentialmedicines/EML_2017_ExecutiveSummary.pdf

Table 2. Some questions are very broad (first, last) so the results are of low interest.

Response: Thanks so much for your observation. As the respondents were registered veterinarians and had core academic understandings on these areas, we believed asking these broad questions would add value while investigating their knowledge on AMR, mechanisms and different class and generations.

Points 3.2 to 3.6 and table 2 to 5. Separate results for livestock and poultry veterinarians should be presented or, at least, differences or not between them must be stated.

Response: Thanks so much for your suggestions. In Bangladesh, the majority of the vets practice in both poultry and livestock sector simultaneously. It was also verified while collecting the list of vets from Bangladesh veterinary council. However, a very few vets are working either in poultry or livestock sector specifically. As it is found that almost all vets are providing support to both livestock and poultry sectors, as a result, it is not possible for us to show vets’ nature wise disaggregated analysis. But we definitely agree with your suggestion.

Table 6. It is not clear how the logistic regression analysis is performed. Please, explain.

Response: Thanks so much for your suggestion. We have explained the process in the data analysis section. Line 187 to 194.

Line 323. I do not see any question regarding signs. In addition, animals have not symptoms.

Response: Thanks for your question. This information came up from the figure 2. We have also dropped ‘symptoms’ from the revised version, throughout.

Line 325. “Better” than...?

Response: Thanks so much for your comment. Please note, we did not want to make any comparison, rather we wanted to make a comment based on the results of figure 2.

Lines 329-330. I do not see this data on the results section. Please, add before discussing.

Response: Thanks for your comment. The related information retrieved from figure 2.  

Line 332. “Infield”?

Response: Deleted.

Lines 342-343. “Confounding factors” are not been analysed in this work. Please, delete this sentence.

Response: Thanks so much for your comment. We did not explore these confounding factors in the current literature. However, as these factors are also related to our study, we wanted to make a commentary, possible reasons of not performing AST. The text has been updated based on literature. Please check the revised version.

Lines 349-350. No sense.

Response: Thanks so much for your comment. We did not explore these confounding factors in the current study. However, we wanted to make a commentary, possible impact of not performing AST. The text has been updated based on literature. Please check the revised version.

Line 353. Please explain what others stakeholders.

Response: Specified.

Lines 354 and 379.   - “Good” is not adequate. A proportion could be high, medium, low, etc.

Response: Thanks so much for your suggestion. Replaced with higher.

Lines 366-368. I do not agree with these sentences regarding your results.

Response: Thanks so much for your comment. We have replaced “access to information” with “KAP”

Lines 376-378. No sense.

Response: Thanks so much for your comment. The relevant information was found from the practice related questions. However, the lines have modified. Please check line numbers 391-394 in the revised version.

Line 386. “Shortage”?

Response: Thanks so much for your comment. We have now changed “shortage” to “lack of”. Please check line number 402 in the revised version.

Lines 389-390. Is this prescription without visiting the farms?

Response: Thanks so much for your question. We have now added “without visiting the farms” in the revised version. Please check line number 405-406 in the revised version.

Line 407. “Other animal health workers”? This manuscript only focus on vets.

Response: Thanks so much for your question. Referring to other studies, we meant that intervention aiming at providing training and education to animal health workers and vets might influence on improved practices to reduce inappropriate use.

Line 410. “Aetilogial factors”? In addition, not use of AST and inability for interpretation should be separated.

Response: Thank you for your suggestion. Firstly, it was a typo on spelling “etiological”. We have changed this sentence like “etiological factors such as not use of antimicrobial susceptibility test and inability for interpreting such reports”. Please see the changes on line number 428-429 in the revised version.

Round 2

Reviewer 1 Report

The authors have addressed the comments and answered the questions. The article can be published as is